

# Ecotoxicological effects of tungsten on celery (*Apium graveolens L*) and pepper (*Capsicum spp.*)

Qi Li[1,2,3,4], Xiaojun Zheng[1,2,3] and Ming Chen[1,2,3]

[1] School of Resources and Environmental Engineering, Jiangxi University of Science and Technology, Ganzhou, Jiangxi, China
[2] Jiangxi Provincial Key Laboratory of Environmental Pollution Prevention and Control in Mining and Metallurgy, Ganzhou, China
[3] Cooperative Innovation Center jointly established by the Ministry and the Ministry of Rare Earth Resources Development and Utilization, Ganzhou, China
[4] Jiangxi Environmental Engineering Vocational College, Ganzhou, China

## ABSTRACT

**Background**. Tungsten (W) is an emerging heavy metal pollutant, yet research remains scarce on the biomonitor and sensitive biomarkers for W contamination.

**Methods**. In this study, celery and pepper were chosen as study subjects and subjected to exposure cultivation in solutions with five different levels of W. The physiological and biochemical toxicities of W on these two plants were systematically analyzed. The feasibility of utilizing celery and pepper as biomonitor organisms for W contamination was explored and indicative biomarkers were screened.

**Results**. The results indicated that W could inhibit plants' root length, shoot height, and fresh weight while concurrently promoting membrane lipid peroxidation. Additionally, W enhanced the activities of superoxide dismutase (SOD), catalase (CAT), peroxidase (POD), and total antioxidant capacity (TAOC) to counteract oxidative damage. From a physiological perspective, pepper exhibited potential as a biomonitor for W contamination. Biochemical indicators suggested that SOD could serve as a sensitive biomarker for W in celery, while TAOC and POD were more suitable for the roots and leaves of pepper. In conclusion, our study investigated the toxic effects of W on celery and pepper, contributing to the understanding of W's environmental toxicity. Furthermore, it provided insights for selecting biomonitor organisms and sensitive biomarkers for W contamination.

## INTRODUCTION

Heavy metals has posed a severe environmental risk (*Topaldemir et al., 2022*; *Yüksel et al., 2022*). Tungsten (W) is a heavy metal widely utilized in daily life and the technological industry. The widespread use of W has increased its environmental risks. W was considered a metal with no significant ecological toxicity or environmental effects for a considerable period, and its ecological and environmental safety was primarily overlooked (*Chen et al., 2019*). Compared to other metallic substances, research on the potential health and

Corresponding author
Ming Chen, jxlgcm@163.com

toxicological effects of W and its compounds on humans has been limited. The United States Environmental Protection Agency (USEPA) officially classified W as an emerging environmental pollutant in 2008, and Russia has similarly designated W as a water pollutant (*Shah et al., 2013*). Although a definitive link between childhood leukemia and the intake of W in this demographic has not been confirmed to date, experimental studies have indicated the toxicity and carcinogenicity of W, particularly its strong mobility in neutral to slightly alkaline water environments (*Du et al., 2020*; *Kelly et al., 2013*; *Lu et al., 2022*). In summary, W has emerged as a novel pollutant currently under investigation. In aquatic environments, W primarily exists in the form of tungstate salts (*Sun & Bostick, 2015*). Plants readily absorb tungstate salts in water and subsequently participate in enzyme synthesis within the plant tissues, and the W may enter the human body through the food chain, leading to health risks (*Preiner et al., 2019*). Therefore, we need to study the process of W transport to plants.

Tailings ponds constitute a major source of W in the environment, and our previous research evaluated the W flux released from a selected tungsten tailings pond into the environment (*Zheng et al., 2024*). The results indicated that a specific tungsten tailings pond annually discharged $6.35 \times 10^8$ mg of W into the environment. The above data means that the concentration of W in the tailings leachate can reach up to 3 mg/L, and it may threaten plants around tailings ponds. *James & Wang (2020)* showed that the enrichment coefficient of W in edible parts of several plants was as follows: radish (28.01), spinach (11.35), potato (1.36), lotus root (0.77), corn (0.32). *Du et al. (2022)* reported that the W content of rice produced near the tungsten mine in Hunan (1.12 mg/kg) was 7.7 times higher than the average W content on the market. *Zheng et al. (2020a)* reported that W is a major heavy contaminant in the soil of the surrounding tungsten mine area. The W plays a certain role in the life processes of plants, such as participating in the activity of certain enzymes and serving as a non-essential nutrient element for plants (*Kennedy et al., 2012*). Meanwhile, as with most heavy metals, excessive W concentration harms organisms. The W entering environmental media (surface water, soil, *etc.*) can potentially impact plant seedlings' growth. The seedling stage is a fundamental phase in plant growth and development and is more susceptible to heavy metal stress.

Some studies have reported the physiological and biochemical toxicity of W on plants. *Dawood & Azooz (2020)* investigated the effects of W on the oxidative status and antioxidant responses of cauliflower seedlings. The study revealed that only when the W concentration exceeded 100 mg/L did it induce the accumulation of hydrogen peroxide, superoxide anions, and hydroxyl radicals in cauliflower, leading to significant membrane degradation in lipid peroxidation. *Adamakis, Panteris & Eleftheriou (2014)* reported that W reduced root growth, particularly by inhibiting cell expansion in the elongation zone, so that root hairs emerged closer to the root tip. The stimulatory effect of W on the biosynthesis of carbohydrates, proteins, free amino acids, as well as enzymatic and non-enzymatic antioxidants may play an important role in protecting broccoli plants against W at low levels (*Dawood & Azooz, 2019*). In contrast, the highest concentration-noxious impacts perceived from oxidative damage and membrane integrity deregulation were accompanied by no gain from increment of proline, superoxide dismutase, and glutathione-S-transferase.

W also promotes cell death by inducing depolymerization and disintegration of microtubule arrays in pea root cells (*Adamakis, Panteris & Eleftheriou, 2011*).

The W reserves in Jiangxi Province, China, account for approximately half of the global reserves (*Wang et al., 2022*), making it a hotspot for W pollution (*Hartley et al., 2014*; *Li et al., 2023a*; *Li et al., 2023b*; *Zheng et al., 2020b*). Existing studies have primarily focused on the physiological and biochemical toxicity of W, with a gap in the literature regarding biomonitor organisms and indicative biomarkers for W pollution. Therefore, there is an urgent need for research on biomonitor organisms and indicative biomarkers for W pollution. Celery and pepper are commonly found terrestrial and aquatic plants in Jiangxi Province, and both possess the potential to serve as biomonitor organisms for W pollution. In conclusion, this study selected celery and pepper as research subjects with the aim of (1) analyzing the physiological and biochemical toxicity of W on celery and pepper, (2) exploring the feasibility of celery and pepper as biomonitor organisms for W pollution, and (3) screening indicative biomarkers with potential applications in W pollution monitoring.

## MATERIALS & METHODS

### Plants incubation

Celery and pepper seeds were purchased from the Shouhe Seeds Industry Co., Ltd. (Shandong province, China). Before the experiment, the seeds were kept in a dark and dry environment. For all vegetables, 150 uniform size seeds were selected. The seeds were sterilized with 10% $H_2O_2$ for 20 min, then washed with ultra-pure water for three times. The seeds were then placed in a Petri dish ($d = 90$ mm) with a double layer of filter paper (150 seeds per dish) and wetted with 10 mL of ultra-pure water. The seeds were germinated at a greenhouse chamber for 7 days. The chamber temperature was maintained at 25 °C, with 16 h/per daylighting and the intensity set at 120 $\mu$mol/ ($m^2 \cdot$ s).

Seedlings with the same growth conditions were selected and transplanted into hydroponic solution with different W concentrations. Hydroponic solutions were prepared by sodium W and the W concentration were 0, 10, 20, 40, and 60 mg/L. And the seedings growth condition was consistency with the seeds germination. Nine plants were placed in each treatment. The culture solution was renewed every three days to prevent changes in the form and concentration of substances in the hydroponic solution. Therefore, a total of 10 treatments (two plants at each of the 5 W concentrations) were harvested after14 days of hydroponics cultivation, washed, and left to be measured. Plants were divided into three tissues for the enzyme activities and W concentration measurement: leaves, stems, and roots.

### Growth of plants determination

Each plant's shoot height (SH) and root length (RL) were measured using a ruler. The fresh weight (FW) is obtained by weighing after removing water from the surface with absorbent paper.

### Biochemical indicator determination

To evaluate the physiological changes of vegetables exposed to W, we measured a group of biomarkers commonly used in toxicological research. About 1.000 g of plant material was
homogenated to measure them with 4.0 mL normal saline (0.9% sodium chloride). Then, the homogenate was centrifuged for 10 min at 3,500 rpm, and the supernatant was used for analysis. All the above indicators were measured using the test kits purchased from Nanjing Jiancheng Bioengineering Institute.

Specifically, membrane damage was detected in terms of "lipid peroxidation" using the biomarker of malondialdehyde (MDA) content measured using the test kit (A003-1, Nanjing Jiancheng Bioengineering Institute). Superoxide dismutase (SOD), peroxidase (POD), catalase (CAT), and total antioxidant capacity (TAOC) were measured using the test kits of A001-1, A084-3-1, A007-1-1, and A015-1 (Nanjing Jiancheng Bioengineering Institute), respectively.

## Pigments (chlorophyll 'a', 'b' and total chlorophyll)

The concentration of chlorophyll 'a' ($C\_a$) and chlorophyll 'b' ($C\_b$) was measured by following the standard acetone-ethanol mixed extraction method. The 0.5 g fresh sample (different tissue) was crushed and the contents were mixed with 10 ml of 80% acetone and preserved in the refrigerator for 5 h to allow complete digestion of chlorophyll pigments in the dark. The filtrate was prepared for analysis with a spectrophotometer at 663 and 645 nm wavelength for determination of chlorophyll a, b, (mg/g fresh weight), the sum of $C\_a$ and $C\_b$ represented the total chlorophyll as follows.

$$C\_a = (12.21 \times A_{663}) - (2.81 \times A_{645}) \tag{1}$$

$$C\_b = (20.13 \times A_{645}) - (5.03 \times A_{663}) \tag{2}$$

$$\text{Total chlorophyll} = C\_a + C\_b \tag{3}$$

where $A_{646}$ and $A_{663}$ are the absorbance of the sample at 645 and 663 nm wavelengths. Reference materials (ASB-00003459-00A and SHAM_105119; Weiye Metrological Reference Materials Research Center, China) were used for quality control with a less than 5% deviation.

## Integrated biomarker response (IBRv2) index

To comprehensively assess the physiological and biochemical parameters in response to tungsten exposure, the Integrated Biological Response version 2 (IBRv2) index was employed. This approach aids in the identification and screening of sensitive biomarkers for W contamination. The calculation of the IBRv2 index follows the methodology outlined in existing literature (*Li et al., 2023a*; *Li et al., 2023b*). The calculation of IBRv2 is as follows.

$$Y_i = \log(X_i/X_0) \tag{4}$$

$$Z_i = (Y_i - X_0)/\sigma \tag{5}$$

$$A_i = Z_i - Z_0 \tag{6}$$

where $X_i$ is the mean of the individual biomarker data, $X_0$ is the mean of the reference data, $Y_i$ is the standardized biomarker response, $Z_i$ is the mean of standardized biomarker response, $\mu$ represents the general mean of $Y_i$, $\sigma$ is the deviation of $Y_i$, $A_i$ is the biomarker deviation index; $Z_0$ is the mean of the reference biomarker data. The sum of the absolute values of the $A_i$ calculated for each biomarker in each treatment studied:

$$IBRv2 = \sum_{i=1}^{n} |Ai| \qquad (7)$$

where the n is the concentration gradient of W ($n = 5$).

## Statistical analysis

All measurements were performed with three parallel samples in all case, the results are expressed in terms of mean and standard deviation (sd). One-way analysis of variance (ANOVA, $p < 0.05$) was used to determine the significant differences among treatments, and then LSD test was performed. Pearson correlation analysis were performed to analyze the correlation among morphophysiological indicators with Origin 2021 and the "ggplot2" package in R (version 4.1.1).

# RESULTS

## Effect of W of celery and pepper growth

The evaluation of plant root or shoot elongation is a widely employed method in environmental biomonitoring (*Wang & Williams, 1990*). Celery and pepper, due to their sensitivity and vitality, are commonly utilized to assess the toxicity of hazardous compounds (*Altaf et al., 2022*; *Long et al., 2003*; *Mumtaz et al., 2022*; *Scoccianti et al., 2006*). In this study, significant ($p \leq 0.05$) linear relationships were observed between the inhibitory effects (%) on root and shoot elongation and the concentration of W. Table 1 outlines the effects of different W concentrations on the growth of celery and pepper. As the W concentration increased, the RL and SH of celery were inhibited. Notably, a stimulating effect on celery RL was observed at a 10 mg/L W concentration. In contrast, no such effect was observed for either RL and SH in pepper at any W concentration. When the W concentration reached 40 mg/L, a noticeable inhibition rate on RL was observed, with 27.6% and 43.7% for celery and pepper, respectively. Pepper's RL was more sensitive to W concentration, with the inhibition rate slightly increasing as the concentration increased. Across the range of 0 to 40 mg/L, the SH of both plants was significantly inhibited (28.6% for celery and 25.1% for pepper). While the inhibition rate for celery's SH remained relatively stable beyond 40 mg/L, the inhibition rate for pepper increased with higher concentrations, indicating greater sensitivity of pepper's SH to W. W exhibited a low-promotion-high-inhibition effect on the FW of both plants. The inflection point for celery occurred at a concentration of 10 mg/L, beyond which inhibitory effects were observed, whereas the inflection point for pepper was at 20 mg/L. These results suggest that pepper may exhibit greater tolerance to W compared to celery. Furthermore, based on the SH indicator, both plants can serve as biomonitor organisms for W pollution, with pepper demonstrating a broader applicability range.
**Table 1  The effects of W concentration on growth of celery and pepper.**

| Vegetable | W conc. (mg/kg) | RL (mm) | SH (mm) | FW (g/plant) | | |
|-----------|-----------------|---------|---------|------|------|------|
| | | | | Root | Stem | Leaf |
| Celery | 0 | 73.41 ± 13.00a | 168.83 ± 21.50a | 0.10 ± 0.02d | 0.91 ± 0.12c | 0.54 ± 0.02c |
| | 10 | 76.33 ± 12.34a | 159.37 ± 23.36b | 0.16 ± 0.03c | 1.16 ± 0.08a | 0.81 ± 0.10a |
| | 20 | 62.93 ± 9.34b | 145.00 ± 19.38c | 0.23 ± 0.05b | 0.98 ± 0.04b | 0.75 ± 0.06b |
| | 40 | 53.12 ± 9.05c | 120.53 ± 13.25d | 0.27 ± 0.03a | 0.40 ± 0.02d | 0.48 ± 0.04d |
| | 60 | 47.23 ± 5.11d | 118.00 ± 20.10e | 0.08 ± 0.01e | 0.29 ± 0.02e | 0.27 ± 0.01e |
| Pepper | 0 | 90.97 ± 10.04a | 155.90 ± 16.95a | 0.72 ± 0.06a | 0.91 ± 0.04b | 1.51 ± 0.07c |
| | 10 | 65.90 ± 10.46b | 142.44 ± 23.41b | 0.42 ± 0.02d | 0.68 ± 0.08c | 1.58 ± 0.06b |
| | 20 | 53.38 ± 10.04c | 130.05 ± 13.12c | 0.63 ± 0.01b | 1.12 ± 0.10a | 1.86 ± 0.04a |
| | 40 | 51.18 ± 8.08d | 116.83 ± 21.93d | 0.53 ± 0.02c | 0.64 ± 0.02d | 1.13 ± 0.03d |
| | 60 | 44.92 ± 5.89e | 81.28 ± 10.21e | 0.34 ± 0.02e | 0.34 ± 0.03e | 0.71 ± 0.06e |

**Notes.**
The different letters show significant difference among different treatments ($P < 0.05$).

The dissolution of W is related to the decrease in oxygen availability and hydrogen ion concentration in water and/or soil, which may be the reason for delayed seedling growth caused by tungstate at plant toxicity levels (*Strigul et al., 2005*). Like most heavy metals, *Adamakis, Panteris & Eleftheriou (2010)* pointed out that W regulates cell formation and elongation by disrupting cortical microtubules, thereby inhibiting root growth. Celery and Pepper have not been reported as accumulators of W. Therefore, W may oxidize proteins in these plants, denature some important enzymes, and affect reactions related to substituting essential metal ions from biological components (*Ghori et al., 2019*). These changes trigger the loss of membrane integrity by altering basic plant metabolic pathways and stimulating the production of reactive oxygen species (*Sallam et al., 2019*).

## Effect of W on pigments

Figure 1 illustrates the impact of W concentration on the chlorophyll content of both celery and pepper, including C_a and C_b. At a W concentration of 0 mg/L, the C_a and C_b content for celery were 0.45 and 0.29 mg/g, respectively, while for pepper, these values were 0.82 and 1.98 mg/g, respectively. For celery, W concentrations ranging from 0 to 40 mg/L had minimal effects on C_a and C_b. However, at a W concentration of 60 mg/L, both C_a and C_b were suppressed, exhibiting reductions of 48.9% and 72.4%, respectively, compared to the 0 mg/L condition.

In the case of pepper, the impact of W concentration on C_a was negligible, and C_a increased with increasing W concentration (with no significant difference) from 0 to 20 mg/L. For C_b in pepper, W exhibited a promoting effect at concentrations up to 20 mg/L; however, further increases in concentration led to the inhibitory impacts, following a trend similar to the changes in pepper's FW. Thus, the influence of W concentration on C_b may play a determining role in pepper FW. The inhibitory effect of W on pigments in celery plants was stronger than that in pepper, influenced by plant characteristics, where

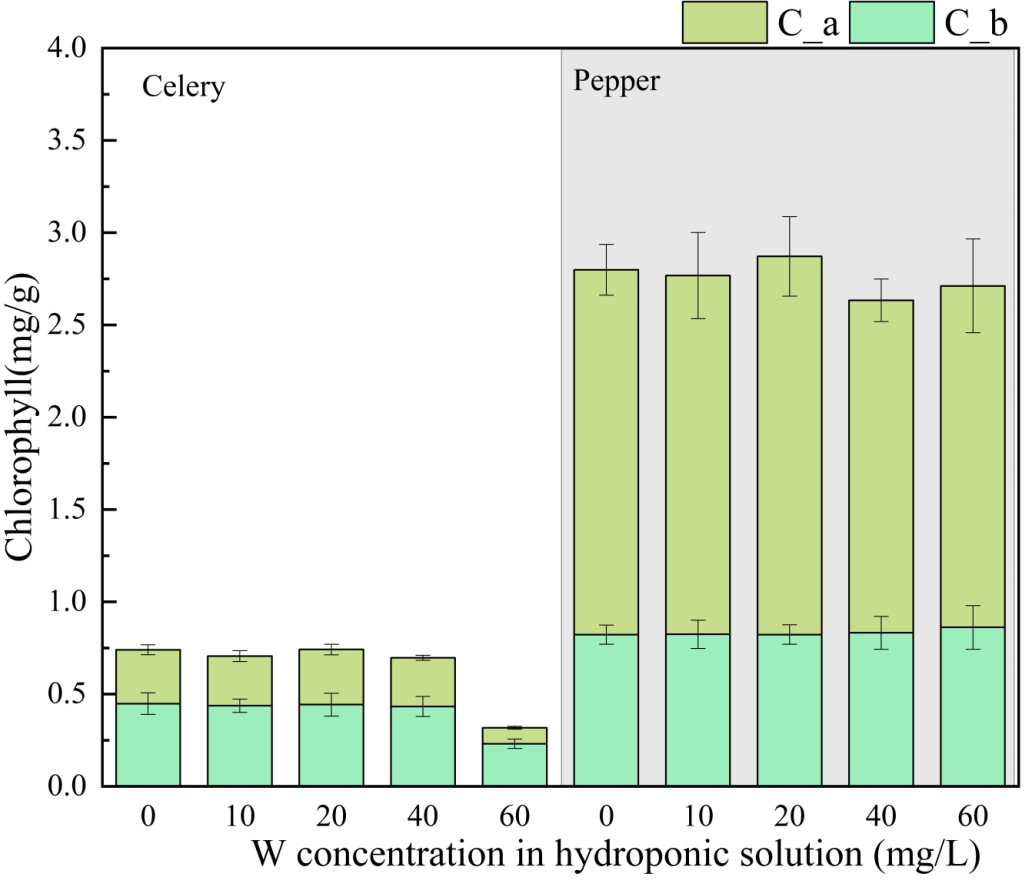

**Figure 1** The effects of W concentration on chlorophyll a, chlorophyll b and total chlorophyll pigments for celery and pepper.

the larger FW of pepper may have a significant role. In conclusion, W exerts a negative or negligible impact on the pigments' content in plants, consistent with previous studies.

## Effect of W on active substances (MDA)

Figure 2 depicts the changes in MDA content in various plant organs (root, stem, and leaf) under W stress. Compared to the control group (0 mg/L), cultivation in 60 mg/L W resulted in an increase of 67.3%, 225.8%, and 292.9% in MDA content for celery roots, stems, and leaves. For pepper, the MDA content in roots stems, and leaves increased by 229.1%, 120.5%, and 40.1%, respectively. MDA content reflects the degree of lipid peroxidation within tissues, indirectly indicating the extent of cellular damage (*He et al., 2024*). It can be inferred that W stress on celery roots is relatively low, while stress on leaves is significant. Conversely, W stress on pepper roots is more substantial, while stress on leaves is less pronounced. The changes in chlorophyll content in Fig. 1 also support the notion that celery leaves are more susceptible to W stress.

Oxidative damage is considered an important cellular effect and has become a commonly used technique for evaluating and comparing the toxicity of various pollutants. The W

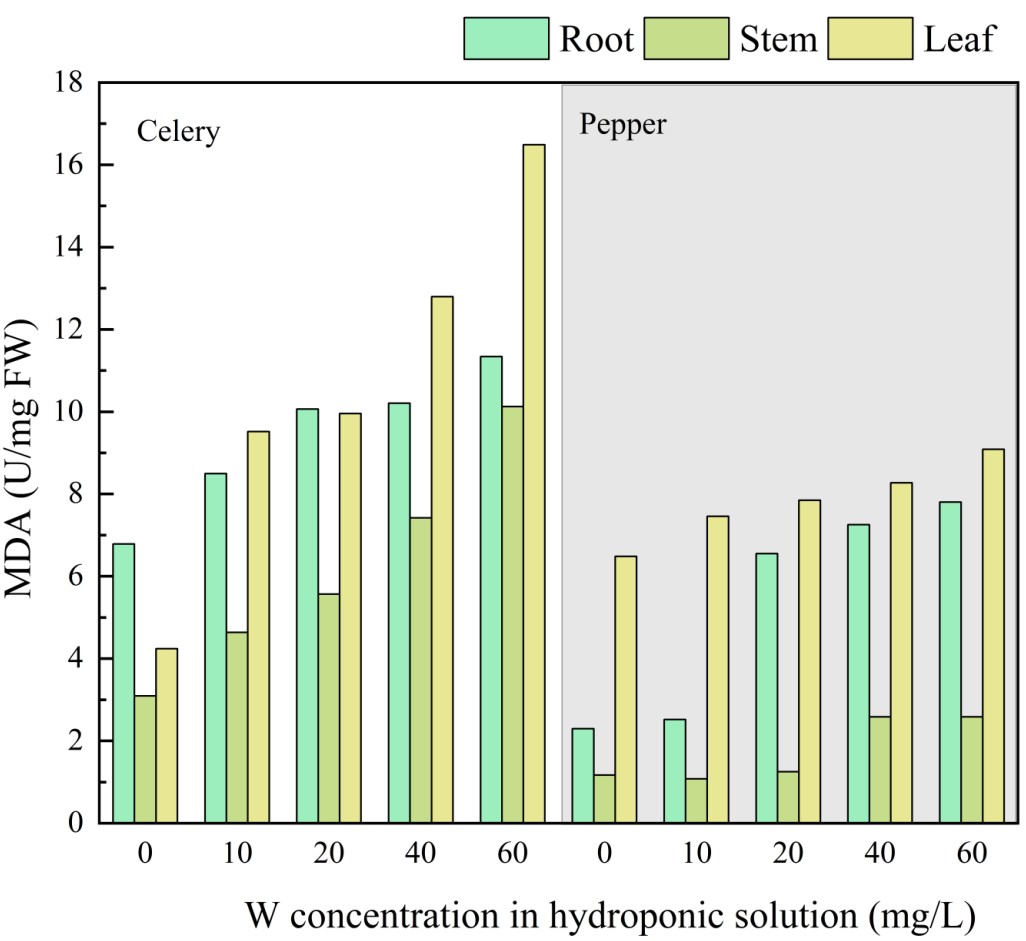

**Figure 2 The effects of W concentration on MDA of the root, stem, and leaf for celery and pepper.**

may induce oxidative stress and accumulate reactive oxygen species in the plant. In this study, the reactive oxygen species of plants were not measured, but the content of MDA can reflect changes in reactive oxygen species. When exposed to environmental stressors, plants secrete active substances such as proline, proteins, soluble sugars, MDA, *etc.*, to increase cell osmotic pressure and enhance tolerance to stressors (*Fang, Hou & Liang, 2021*). In this study, the MDA content in plant organs increased with the rise in W concentration, but the magnitude of the increase varied between different plants and organs of the same plant. Both plants exhibited the lowest stem MDA content, attributed to their classification as vascular plants with a tough outer layer (bark) and a woody interior (xylem). Research indicates that plant stem tissues primarily transport nutrients and water, making them less susceptible to W stress.

## Effect of W on the antioxidative defense system (CAT, SOD, POD and TAOC)

In this study, the antioxidant system of two plants under different W concentrations was investigated by analyzing SOD, CAT, POD, and TAOC in various plant organs (Fig. 3).

**Table 2  The correlationship between W concentration and antioxidative defense system.**

| Plant | Index_Tissues | Correlation | Index_Tissues | Correlation |
|---|---|---|---|---|
| Celery | SOD_Root | 0.86 | POD_Root | 0.93 |
| | SOD_Stem | 0.89[*] | POD_Stem | 0.97[*] |
| | SOD_Leaf | 0.88[*] | POD_Leaf | 0.91[*] |
| | CAT_Root | 0.87 | TAOC_Root | 0.91[*] |
| | CAT_Stem | 0.96[*] | TAOC_Stem | 0.89[*] |
| | CAT_Leaf | 0.96[**] | TAOC_Leaf | 0.95[*] |
| Pepper | SOD_Root | 0.93[*] | POD_Root | 0.96[*] |
| | SOD_Stem | 0.94[*] | POD_Stem | 0.90[*] |
| | SOD_Leaf | 0.97[**] | POD_Leaf | 0.80 |
| | CAT_Root | 0.86 | TAOC_Root | 0.92[*] |
| | CAT_Stem | 0.96[*] | TAOC_Stem | 0.99[*] |
| | CAT_Leaf | 0.91[*] | TAOC_Leaf | 0.96[**] |

**Notes.**
[*]Indicating significance at the $p < 0.05$ level.
[**]Indicating significance at the $p < 0.01$ level.

Compared to the control group, W stress increased the SOD content in celery roots, stems, and leaves by 360.6–521.1%, 96.8–186.6%, and 152.9–248.7%, respectively. For pepper, the respective increments in SOD content in different tissues were 1.8–51.8%, 23.1–57.7%, and 5.9–20.4%. The POD content in celery tissues showed an increment ranging from 5.9% to 202.8%, while in pepper, the increment ranged from 14.8% to 157.6%. The increment of CAT content in celery tissues ranged from 61.2% to 2645.6%, while in pepper, the increment ranged from 9.1% to 890.0%.

Plants subjected to heavy metals or organic pollutants may generate reactive oxygen species such as hydrogen peroxide ($H_2O_2$), superoxide anion radical ($\cdot O_2^-$), and hydroxyl radical (·OH) (*Guo et al., 2022*; *Gao et al., 2022*). To counteract the cellular damage caused by reactive oxygen species and maintain the homeostatic balance of plant cells, the defense system of the plant organism can eliminate reactive oxygen species through the production of enzymes (including CAT, POD, and SOD, *etc.*) or non-enzymatic antioxidants (such as glutathione and hydroxybenzene, *etc.*) (*Jan et al., 2023*). The study indicates that plants counteract heavy metal stress by increasing SOD, POD, and CAT to eliminate excess reactive oxygen species (*Zhou, Wang & Inyang, 2021*). Additionally, TAOC reflects the sum of the antioxidant capacity of enzymes and non-enzymatic components (*Huang et al., 2023*). As shown in Fig. 3D, the TAOC content in plant tissues is positively correlated with W concentration. Table 2 presents the correlation analysis between the antioxidative defense system in plant organs and W stress. The results indicate that the antioxidant capacity of enzymes in celery roots is not significantly correlated with W concentration, which may be attributed to the promoting effect of low concentration W on celery (Table 1). While, antioxidant capacity of enzymes in pepper roots has a significant correlation ship with W concentration. The antioxidant capacity of enzymes in pepper roots may be a primary contributor to pepper's tolerance to W stress.

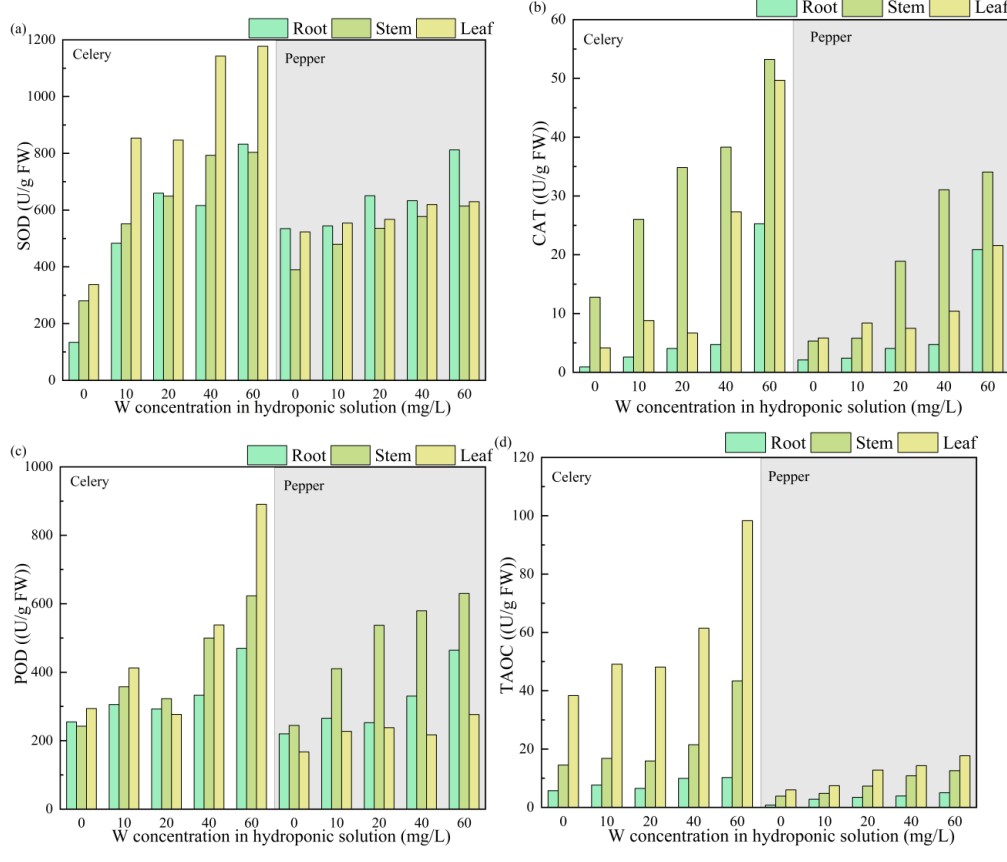

**Figure 3** The effects of W concentration on (A) SOD, (B) CAT, (C) POD and (D) TAOC of the root, stem, and leaf for celery and pepper.

## Evaluation of comprehensive phytotoxicity of W

The IBRv2 index combines the biological effects of multiple biomarkers and is used to screen for sensitive biomarkers. In this study, data from five biomarkers (MDA, SOD, POD, CAT, and TAOC) were standardized to create a radar chart (Fig. 4). It is important to note that in this section, our primary focus is on the roots and leaves of the plants. The baseline (depicted in red) represents the control group, and when the index is greater than 0, it indicates biomarker activation. All five biomarkers in the roots are activated for celery except for POD at a W concentration of 20, suggesting a deactivation. In the case of pepper, W at 10 mg/L does not significantly activate MDA, SOD, CAT, and POD in the roots, possibly due to pepper's tolerance to low concentrations of W. Figs. 4E, 4F present the IBRv2 values for the five indicators. Generally, the indicator with the highest IBRv2 value is selected as the sensitive biomarker. The maximum IBRv2 values for celery roots and leaves are observed in SOD, while for pepper roots and leaves, they are observed in TAOC and POD, respectively. The levels of these antioxidant enzymes (SOD and POD) are crucial indicators in plant toxicity tests, closely related to the degree of oxidative damage. They are important markers for assessing the impact of environmental pollutants

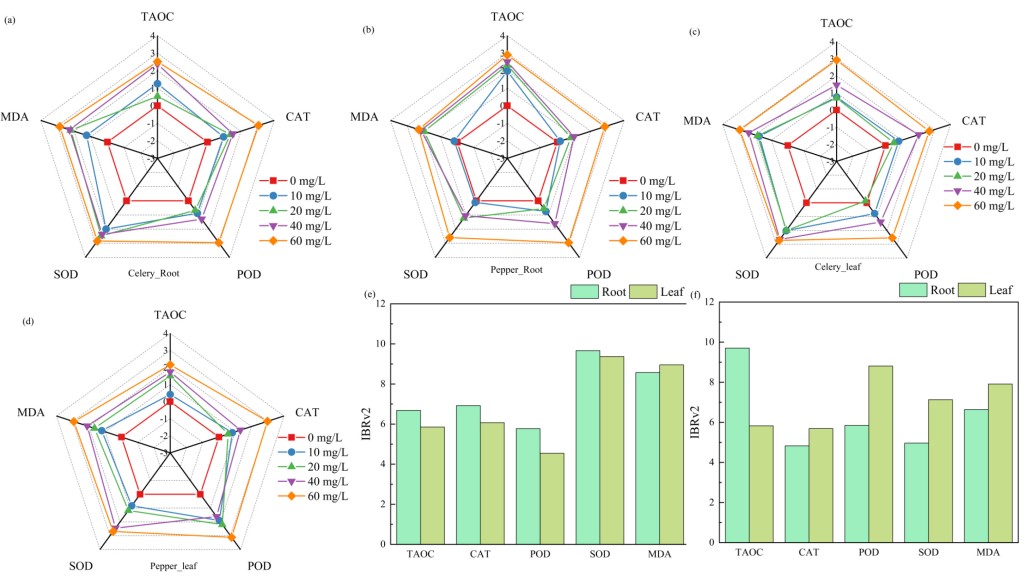

**Figure 4** Star plots for the IBRv2 index of (A) celery root, (B) pepper root, (C) celery leaf, and (D) pepper biomarkers exposed to W. And IBRv2 index values of (E) celery and (F) pepper under W treatments.

on plant growth (*Zhang et al., 2018*). TAOC represents the activity of the non-enzymatic antioxidant defense system. Our results reflect the potentially toxic effects of W on the celery and pepper, which complements research on the celery and pepper as bio-indicator.

## CONCLUSIONS

This study investigated the ecotoxicological impact of W on celery and pepper at the physiological and biochemical levels. Low concentrations of W were found to enhance plant biomass, while high concentrations had the opposite effect. The different inflection points in the concentration–response curves for both plants suggest that pepper exhibits tolerance to W. This conclusion was further supported by pigment and MDA analyses, emphasizing the greater potential of pepper as a biomonitor compared to celery. W was observed to inhibit plant growth by reducing pigments, with $C_b$ being more affected than $C_a$. Additionally, W induced oxidative damage and membrane lipid peroxidation, thereby impacting the growth of both celery and pepper. Furthermore, the IBRv2 index analysis identified SOD as a sensitive biomarker for W in celery, while TAOC and POD were deemed more suitable for sensitive biomarkers in the roots and leaves of pepper. In summary, our research delved into the toxic effects of W on celery and pepper, complementing the understanding of W's environmental toxicity. The findings provide valuable insights for selecting biomonitors and sensitive biomarkers in W pollution. In addition, the toxicity test conducted in this study was short-term (14 days), and the plants did not mature and produce. Considering food security and human health, it is necessary to conduct research

on crops under long-term exposure to W in the future, and the accumulation of W in plant organs should be given more attention.

### Funding
This work was funded by the National Key R&D Program of China (No. 2019YFC1805100), the National Natural Science Foundation of China (No. 51664025), the Jiangxi Provincial Natural Science Foundation (No. 20232ACB203026), and the Science and Technology Project of Ganzhou City (No. 2023PNS27982), the Science and Technology Project of Jiangxi Education Department (No. GJJ214407). The funders had no role in study design, data collection and analysis, decision to publish, or preparation of the manuscript.

### Grant Disclosures
The following grant information was disclosed by the authors:
The National Key R&D Program of China: No. 2019YFC1805100.
The National Natural Science Foundation of China: No. 51664025.
The Jiangxi Provincial Natural Science Foundation: No. 20232ACB203026.
The Science and Technology Project of Ganzhou City: No. 2023PNS27982.
The Science and Technology Project of Jiangxi Education Department: No. GJJ214407.

### Competing Interests
The authors declare there are no competing interests.

### Author Contributions
- Qi Li conceived and designed the experiments, performed the experiments, prepared figures and/or tables, authored or reviewed drafts of the article, and approved the final draft.
- Xiaojun Zheng analyzed the data, prepared figures and/or tables, and approved the final draft.
- Ming Chen analyzed the data, prepared figures and/or tables, and approved the final draft.

### Data Availability
The raw data are available in the Supplementary File.

### Supplemental Information
Supplemental information for this article can be found online at http://dx.doi.org/10.7717/peerj.17601#supplemental-information.

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
