# Peer review of "Ecotoxicological effects of tungsten on celery (Apium graveolens L) and pepper (Capsicum spp.)"

_PeerJ, doi:10.7717/peerj.17601_

## Round 0.1 · original submission · Major Revisions

Dear authors, please follow the reviewers suggestions, specifically make sure to clear the experimental details and data presentation, improving their interpretation and discussion.

Reviewer 3 has requested that you cite specific references. You may add them if you believe they are especially relevant. However, I do not expect you to include these citations, and if you do not include them, this will not influence my decision.

**Language Note:** The review process has identified that the English language must be improved. PeerJ can provide language editing services - please contact us at [email protected] for pricing (be sure to provide your manuscript number and title). Alternatively, you should make your own arrangements to improve the language quality and provide details in your response letter. – PeerJ Staff

Reviewer 1 ·

Basic reporting

Your introduction needs more detail. I suggest that you improve the description at lines 68-83 to provide more justification for your study (specifically, you should expand upon the knowledge about mechanism of tungsten contamination of plants).

Experimental design

no comment

Validity of the findings

no comment

Reviewer 2 ·

Basic reporting

The manuscript investigates the ecotoxicological effects of tungsten on celery and pepper plants, focusing on biomonitoring and sensitive biomarkers for tungsten contamination. It highlights tungsten's emerging status as an environmental pollutant, underscoring its potential health implications. The study explores growth, biochemical responses, and pigment changes in plants, emphasizing the need for biomonitor organisms and indicative biomarkers. However, improvements in clarity regarding experimental details, data presentation, and interpretation could enhance the overall impact and reproducibility of the study.
In the introduction section:
• add a sentence or two for a smoother transition between tungsten's role in plants and its emerging environmental pollutant status.
• Clarify the link between tungsten and potential human health effects, like childhood leukaemia, and explain its relevance to the study.
• Emphasize why the tungsten flux data is crucial to the study. Connect it to the environmental impact and the rationale behind studying its effects on plant seedlings.

Experimental design

In the method section:
• specify the conditions during the 7-day incubation period, such as temperature and light intensity. This information is important for reproducibility.
• Provide the units of W concentration used
• if the 10 treatments mentioned include both plant types (celery and pepper) at each of the 5 W concentrations or if it refers to a different experimental setup
• why the plants were harvested after 14 days of hydroponic cultivation?

Validity of the findings

• How were chlorophyll and carotenoid concentrations measured? Include specific extraction methods and any deviations from the standard protocol.
• Elaborate on why certain concentrations led to suppression, and how this relates to the overall health and function of the plants.
• how the IBRv2 index values were calculated and interpreted. Provide a more detailed discussion on the overall phytotoxicity evaluation and its implications for each biomarker.

Additional comments

Check subscripts and superscripts in units, chemical formulas, etc.
Enhance the caption of figures by making them more elaborative
Check correct use of acronyms, grammar, spacing

Reviewer 3 ·

Basic reporting

Ecotoxicological effects of tungsten on celery (Apium graveolens L) and pepper (Capsicum spp.)

General Information


The study investigates the impact of tungsten on celery and pepper plants, analyzing physiological and biochemical toxicities, potential biomonitoring capabilities, and indicative biomarkers for tungsten contamination.

The ecotoxicological effects of tungsten on celery (Apium graveolens L.) and pepper (Capsicum spp.) represent an important area of study due to the potential risks posed to agricultural ecosystems and human health. Tungsten, a heavy metal widely used in industrial processes, can contaminate soil and water sources, leading to uptake by plants and subsequent bioaccumulation in the food chain. Research into the effects of tungsten exposure on celery and pepper plants is critical for assessing the impact on crop growth, physiological processes, and biochemical responses. Understanding how tungsten influences these key parameters can provide valuable insights into the potential risks associated with its presence in agricultural environments, as well as inform strategies for mitigating contamination and safeguarding food safety. Therefore, investigating the ecotoxicological effects of tungsten on celery and pepper plants contributes to our broader understanding of metal-induced stress in agricultural ecosystems and aids in the development of sustainable farming practices.

Strengths

• Extensive data set compiled over years of detailed fieldwork.
• Clear and professional language used throughout the manuscript.
• The study addresses an important gap in the literature by investigating the ecotoxicological effects of tungsten on two commonly consumed crops, celery, and pepper.
• Understanding the potential impact of tungsten contamination on agricultural crops is crucial for assessing food safety and environmental health.
• The experimental design appears thorough, incorporating both greenhouse and laboratory studies to assess the effects of tungsten exposure on plant growth, physiological parameters, and biochemical responses.


Deficiencies

• Descriptive metadata identifiers for supplemental files could be enhanced.
• The introduction should provide more background information on the sources of tungsten contamination in agricultural environments and its potential ecological and human health implications.
• The methods section lacks detailed information on key aspects such as the concentrations of tungsten used, exposure duration, and statistical analyses employed.
• The results section needs to provide clearer interpretations of the findings, including the significance of observed effects on plant growth, physiology, and biochemistry.


Critical Questions

• How was the statistical analysis conducted, and are there any plans for further refinement?
• Can the authors provide more detailed metadata identifiers for supplemental files?
• What specific improvements can be made to enhance the manuscript's statistical analysis?
• Can the authors provide more details on the concentrations of tungsten used in the experiments and how these relate to environmentally relevant levels of tungsten contamination in agricultural soils?
• Do the authors have any hypotheses regarding the underlying mechanisms through which tungsten exposure affects the physiological and biochemical responses observed in celery and pepper plants?
• What are the potential implications of the study findings for agricultural practices and food safety regulations, particularly in regions where tungsten contamination may be prevalent?


Recommendations
• Provide more descriptive metadata identifiers for supplemental files.
• Consider addressing the identified deficiencies to strengthen the manuscript. General ecotoxicological health risk assessment should be emphasized. I strongly recommend authors to benefit from following recent studies, citing them within the manuscript.
https://doi.org/10.1007/s11356-022-23937-2
https://doi.org/10.1007/s11356-021-17023-2
• Provide a more comprehensive overview of the environmental sources of tungsten contamination and its potential impacts on agricultural ecosystems and human health.
• Clearly outline the experimental procedures, including details on tungsten concentrations, exposure duration, and statistical analyses performed.
• Ensure that the results section provides thorough explanations of observed effects and their significance for plant health and productivity.
• Consider discussing potential future research directions, such as exploring the long-term effects of tungsten exposure on crop growth and yield, as well as investigating remediation strategies for mitigating tungsten contamination in agricultural soils.

Decision

Consequently, the subject matter exhibits a high level of intrigue, while the research itself demonstrates a thorough and complete design. The key points and methodology were well explained. The paper can be ACCEPTABLE after a minor revision.

Experimental design

It has been stated in basic reporting

Validity of the findings

It has been stated in basic reporting

Additional comments

It has been stated in basic reporting

---

## Round 0.2 · Minor Revisions

The manuscript can be accepted after the very minor modifications requested by the Rev1. Regards

Reviewer 1 ·

Basic reporting

Please check that the expression of numbers in Tables in this manuscript are correct and that scientific notation is recommended.

Experimental design

no comment

Validity of the findings

no comment

Additional comments

This manuscript may be accepted after minor modification.
Please check that the weight of the seedings are correct in this manuscript.

Reviewer 2 ·

Basic reporting

No comment

Experimental design

NO comment

Validity of the findings

no comment

Additional comments

Revision is done appropriately and can be considered for the further publication process

Reviewer 3 ·

Basic reporting

.

Experimental design

.

Validity of the findings

.

Additional comments

The authors have addressed all the questions and recommendations. I believe this work can be acceptable for publication.

---

## Round 0.3 · accepted · Accept

The MS can be accepted, I confirm the authors have addressed all of the reviewers' comments.
Regards,
FC